# Alliin, An *Allium sativum* Nutraceutical, Reduces Metaflammation Markers in DIO Mice

**DOI:** 10.3390/nu12030624

**Published:** 2020-02-27

**Authors:** Marina A. Sánchez-Sánchez, Adelaida Sara Minia Zepeda-Morales, Lucrecia Carrera-Quintanar, Juan Manuel Viveros-Paredes, Noel Noé Franco-Arroyo, Marisol Godínez-Rubí, Daniel Ortuño-Sahagun, Rocío Ivette López-Roa

**Affiliations:** 1Laboratorio de Neuroinmunobiología Molecular, Instituto de Investigación en Ciencias Biomédicas (IICB) CUCS, Universidad de Guadalajara, Guadalajara Jalisco 44340, Mexico; marina.sanchez@alumnos.udg.mx; 2Laboratorio de Investigación y Desarrollo Farmacéutico, CUCEI, Universidad de Guadalajara, Guadalajara Jalisco 44430, Mexico; adelaida.zepeda@academicos.udg.mx (A.S.M.Z.-M.); juan.viveros@academicos.udg.mx (J.M.V.-P.); noel.franco@alumnos.udg.mx (N.N.F.-A.); 3Laboratorio de Ciencias de los Alimentos, Departamento de Reproducción Humana, Crecimiento y Desarrollo Infantil, CUCS, Universidad de Guadalajara, Guadalajara Jalisco 44340, Mexico; lucrecia.carrera@academicos.udg.mx; 4Laboratorio de Investigación en Patología, Departamento de Microbiología y Patología, CUCS, Universidad de Guadalajara, Guadalajara Jalisco 44340, Mexico; juliana.godinez@academicos.udg.mx

**Keywords:** obesity, inflammation, cytokines, adipocytokines, s-allyl cysteine sulfoxide, adipose tissue

## Abstract

Obesity generates a chronic low-grade inflammatory state which promotes oxidative stress and triggers comorbidities. Alliin is the main organosulfur compound in garlic and has been shown to induce a decrease in the expression of proinflammatory cytokines; its systemic effect on metabolic parameters and adipose tissue is not yet known, however. After nine weeks of HFD and with obesity established in C57BL/6 mice, we observed that a daily treatment with alliin for 3.5 weeks (15 mg/kg) did not affect body weight, but significantly improved insulin sensitivity and glucose tolerance, both evaluated through a blood glucose monitoring system. Once alliin treatment was completed, serum, adipose tissue, and organs of interest related to metabolism were removed for further analysis. We observed that alliin significantly decreased the size of adipocytes from epididymal adipose tissue, evaluated via microscopy. A decrease in gene expression and serum protein levels of the adipocytokines leptin and resistin, as well as decreased serum IL-6 concentration, were detected by qRT-PCR and ELISA, respectively. It did not, however, affect mRNA expression of antioxidant enzymes in the liver. Taken altogether, these results indicate that treatment with alliin reduces metaflammation markers in DIO mice and improves some metabolic parameters without affecting others.

## 1. Introduction

Obesity is the excess of adipose tissue that presents a risk for health. It generates metabolic distress and alterations in the release of hormones and cytokines from the adipocytes and immune cells [1]. Stressed adipose tissue generates a chronic low-grade inflammatory state, characterized by elevated circulatory levels of proinflammatory cytokines, cellular infiltration, and the activation of inflammatory pathways [2]. 

Inflammatory cytokines secreted by adipose tissue and the influence of high Tumor Necrosis Factor α (TNFα) levels on the development of insulin resistance in obesity were first described in a study on obese mice by Hotamisligil et al. in 1993 [3]. Additionally, they noted that the major pathway of the insulin resistance mechanism lies in the alteration of the phosphorylation of insulin receptor substrate proteins (IRS) produced by TNFα after affecting the IκB kinase (IKK) complex, and the Jun kinase (JNK) and protein kinase C (PKC) signaling pathways [2]. The alteration of signaling proteins from inflammatory pathways favors the greater expression of oxidative and nitrosative reactive species which also have a role in the activity of NF-kB, therefore perpetuating the inflammatory process [4]. Oxidative stress caused by caloric excess is described as one of the primary pathogenic processes of diabetes mellitus since it can cause cytotoxicity of β cells in the pancreas [5].

This inflammatory and oxidative environment, derived from metabolic dysfunction and the activation of immune cells, is known as metaflammation [6]. The inflammatory and dysmetabolic states trigger comorbidities related to obesity such as insulin resistance, type 2 diabetes mellitus, and dyslipidemia [7]. Despite knowledge of these mechanisms, current therapeutic and nutritional approaches in fighting obesity have proven to be insufficiently effective. 

The term nutraceutical derives from "nutrition" and "pharmaceutical". It consists of a food (or part of a food) that provides health benefits and that can additionally prevent or treat an illness [8,9]. A large quantity of these bioactive molecules can be found in garlic (*Allium sativum*) [10,11,12]. In particular, alliin, the major organosulfur compound of garlic, has been researched for its antidiabetic, anticarcinogenic, antioxidant, and anti-inflammatory effects [11,13,14].

In 3T3-L1 adipocytes stimulated with lipopolysaccharides (LPS), alliin induces a decrease in the expression of proinflammatory cytokines such as Monocyte Chemoattractant Protein-1 (MCP-1) and Interleukine-6 (IL-6) through the inhibition of phosphorylation of ERK1/2 signaling proteins [15]. In addition, when evaluated in murine models of inflammatory bowel disease, alliin induces a decrease in the expression of cytokines IL-6 and TNFα, besides the significant suppression of the phosphorylation of the signaling pathways ERK1/2, JNK, and p38 [16]. On this basis, the present work focuses on the effect of alliin supplementation on metainflammatory markers for energy metabolism, antioxidant enzymes, and proinflammatory cytokines in diet-induced obesity (DIO) mice.

## 2. Materials and Methods 

### 2.1. Alliin 

Alliin (S-Allyl-L-cysteine sulfoxide) with empirical formula C6H11NO3S was purchased from Sigma-Aldrich (St. Louis, MO, USA) CAS Number: 17795-26-5. The lyophilizate was eluted in physiological saline solution (10 mg/600 µL).

### 2.2. Diets

The standard diet consisted of 3.1 kcal/g: 24% calories from protein, 58% calories from carbohydrates, and 18% calories from fat (Teklad Global 18% Protein Rodent Diet, Envigo. Huntingdon, UK). Saturated high-fat diet composition was 5.1 kcal/g: 18.3% calories from protein, 21.4% calories from carbohydrates, 60.3% calories from fat (TD.06414 Diet, Envigo. Huntingdon, UK).

### 2.3. Animals

Eight-week-old male C57BL/6 mice (~25 g) were purchased from the Universidad Autónoma Metropolitana (Xochimilco, Mexico). This project was approved by the Ethical Committee in Research and Biosecurity of the Centro Universitario de Ciencias de la Salud, Universidad de Guadalajara (Approval CI-012108). Animals were lodged at room temperature ~26 °C in a 12 h daylight cycle and allowed food and water ad libitum, except during tests which required fasting. After one week of acclimatization, mice were randomly separated into two groups: the standard diet group or STD (*n* = 23) and the high-fat diet group or HFD (*n* = 21). After 63 days (9 weeks), the body weight gain was significant and obesity established. We then formed the following four experimental groups: HFD, continuing the high-fat diet (*n* = 10); HFD+A, receiving the high-fat diet and alliin treatment (15 mg/kg) (*n* = 11); STD, fed with standard diet (*n* = 12); and STD+A, who were treated with alliin (15 mg/kg) alongside the standard diet (*n* = 11) (Figure 1). The treatment was administered daily via an oral gavage technique for 25 days (3.5 weeks). Finally, they were euthanized by decapitation with scissors for whole blood collection and tissue dissection.

### 2.4. Metabolic Tests

The metabolic tests were performed at basal time (day 0), after obesogenic diet (day 63), and once the alliin treatment was finished (day 88).

#### 2.4.1. Oral Glucose Tolerance Test (OGTT) and Intraperitoneal Insulin Tolerance Test (IITT)

Mice were previously fasted for 4 h. According to the test, glucose levels were determined at minute 0 and then mice were administered 2 g/kg oral glucose or 0.75 UI/kg intraperitoneal insulin for the OGTT and the IITT, respectively, following the glucose determination at minutes 15, 30, 60, and 120 with an On-Call® Plus Blood Glucose Monitoring System (ACON Laboratories, Inc, USA).

#### 2.4.2. Cholesterol and Triglycerides Test

After 4 h fasting, a single measurement of cholesterol and triglycerides levels was performed with the Accutrend® Plus equipment (Roche Diagnostics, USA).

### 2.5. Serum Cytokines Measurement 

Whole blood was collected in 1.5 ml tubes and incubated for clot formation. After 15 min, samples were centrifugated at 2000 rpm for 10 min. Approximately 50 μL of serum was obtained and stored in aliquots at −80 °C for later determination of serum cytokines: Amylin, C-peptide 2, Ghrelin, GIP, GLP-1, Glucagon, insulin, PP, PPY, TNFα, IL-6, MCP-1, Leptin, and Resistin with Mouse Metabolic Magnetic Bead Panel MILLIPLEX® MAP Cat. MMHMAG-44K and Luminex® MAGPIX® (MAGPIX System, Luminex, USA) according to the supplier.

### 2.6. Histological Examination and Hematoxylin–Eosin Staining

The epididymal adipose tissue was dissected and fixed overnight at room temperature in formaldehyde solution (4 %), dehydrated, and embedded in paraffin. Slides (5 µm thick paraffin sections) were stained with H&E [16], imaged using bright-field microscopy (DM2500 Leica Microsystems GmbH, Germany), and observed with Texas Red Filter. For the automated adipocytes size analysis, CellProfiler software was used [17]. 

### 2.7. Quantitative Real-Time PCR (qRT-PCR) Analysis

Adipose tissue total RNA of eight samples from each group was isolated using RNeasy® Lipid Tissue Mini Kit Cat. #74104 (Qiagen, Manchester, UK). cDNA was synthesized using iScript Adv cDNA kit for RT-qPCR Cat. #1725038 (Bio-Rad, Hercules, CA, USA) according to the manufacturer’s instructions. 

For gene expression, a SYBR green-based, Cat. #172-5270 (Bio-Rad, Santa Rosa, CA, USA), real-time quantitative PCR (RT-qPCR) assay was performed on a Rotor Gene Q (Qiagen, Hilden, Germany) and relative expression ratios were determined by the Rotor-Gene Q Software 2.3.4. 

Primers were designed online with the primer-BLAST tool and are listed in Table 1; GAPDH was used as a housekeeping gene. The procedure was performed with a 10 µl reaction mixture containing 5 ul of SYBR green enzyme, 0.4 ul of each forward and reverse primer, 2.2 ul of water, and 2 ul of cDNA under the following conditions: denaturation at 95 °C for 30 s, 45 cycles at 95 °C for 15 s, annealing temperature (varying for each gene as shown in Table 1) for 15 s, an extension of 72 °C for 30 s only for products longer than 150 pb, and the Melt ramp from 45 °C to 95 °C. Taqman probe was employed for IL-6 (Mm00446190_m1) (Thermo Fisher, Waltham, MA, USA).

### 2.8. Statistical Analysis

D’Angostino–Pearson normality test was applied. One-way ANOVA, Student´s T-test, or the Kruskal–Wallis test was performed, corresponding to data behavior. A *p* ≤ 0.05 was taken as significant and was determined with Tukey–Kramer’s or Dunn’s multiple statistics test using GraphPad Prism 6 software (version 19.0, IBM Inc., Chicago, IL, USA). Gene expression was normalized against the expression level of GAPDH from the control group. Relative gene expression level was calculated using the 2^-∆∆CT^ method. The representative results of each study group were expressed as mean ± standard deviation.

## 3. Results

### 3.1. Alliin Does Not Affect Body Weight but Significatively Improves Glucose Tolerance

Body weight was monitored each week from the beginning of the administration of different diets. The effect of the high-fat diet was observed from day 7 of administration (HFD vs. STD, *p* < 0.0001); this significance oscillated until day 35, after which it remained stable. On the other hand, 25 days of alliin treatment did not show an effect on total body weight in any of the treated groups (Figure 2). 

The metabolic tests performed at the end of the alliin treatment indicated the tendency of alliin to improve the levels of metabolic parameters (Table 2). No statistically significant difference was found in most of them, however, except in the area under the curve in the oral glucose tolerance test, in which the HFD group presented the following statistically significant difference in comparison to the other three groups: with STD group (****p* < 0.001), and with both HFD+A and STD+A groups (**p* < 0.05) (Figure 3B), as well as during the first minutes of the insulin tolerance test (data not shown). The polygonal graph represents the normalized levels obtained in the metabolic tests (Figure 3A).

### 3.2. The Obese Group Treated with Alliin Had Smaller Adipocyte Size Than the Obese Control Group

The weight of the adipose deposit of the epididymis was differently affected in relation to the type of diet administered, but alliin treatment did not show an effect in any of the groups (Table 2). This notwithstanding, at the cellular level, the high-fat diet increased the size of the adipocytes of the HFD group compared to the STD group (*p* < 0.0001). When alliin was administered in the HFD+A group, the size of the adipocytes decreased significantly (HFD+A vs. HFD *p* < 0.0001), although the HFD+A group still presented a greater size than the control groups (both STD or STD+A; *p* < 0.0001) (Figure 4).

### 3.3. Alliin Significantly Decreased Gene Expression from Epididymal Adipose Tissue and Protein Serum Levels of Adipocytokines Leptin and Resistin.

Notoriously, leptin and resistin, adipocytokines directly related to insulin metabolism, are increased by the high-fat diet in a significant way. The effect of the high-fat diet is visible in resistin concentration when comparing the STD and HFD groups and the STD+A and HFD groups (*p* < 0.001) and in the HFD vs. HFD+A groups (Figure 5A). On the other hand, leptin concentration was increased in the HFD group when compared with all others (*p* < 0.0001). Alliin treatment in the HFD+A group significantly decreased leptin levels (HFD+A vs. HFD *p* < 0.0001), although the HFD+A group still presented an elevated concentration when compared with the control groups (STD or STD+A; *p* < 0.0001) (Figure 5C).

The resistin mRNA expression (Figure 5B) showed the same significant difference between the HFD and HFD+A groups (*p* < 0.001) as that observed in the protein secretion. Interestingly, a higher gene expression was observed in the STD+A group compared to the STD group (*p* < 0.05). In fact, STD+A resistin expression was even higher than that observed in the HFD+A group (*p* < 0.0001). An additional difference was observed between the STD+A and HFD groups (*p* < 0.0001).

In the case of leptin mRNA expression (Figure 5D), there was a difference between the control groups of the STD vs. HFD diets (*p* < 0.0001) as well as a greater expression of this gene in the HFD group (*p* < 0.01) compared to STD+A. As with resistin, a statistical difference was found between the HFD and HFD+A groups (*p* < 0.001).

### 3.4. Obesity-Related Serum Inflammatory Cytokine Concentrations Were Decreased or Homogenized by Alliin

The determination of the most reported acute-phase cytokines related to the low-grade inflammatory process in obesity indicates that, indeed, the high-fat diet increased IL-6 serum levels between STD vs. HFD (*p* < 0.001) and STD+A vs. HFD (*p* < 0.01). Meanwhile, alliin demonstrated its anti-inflammatory capacity by, on the one hand, decreasing the levels of IL-6 in the HFD+A group in a significant way (*p* < 0.01) (Figure 6A), and on the other, exhibiting the interesting tendency to homogenize the levels of TNFα and MCP-1 (Figure 6B,C). 

The increment of inflammatory cytokines and the development of a low-grade inflammatory state in obesity is directly associated with the metabolic stress of adipocytes and the recruitment of inflammatory immune cells [6,18,19]. Thus, in this DIO model, we took the epididymal adipose tissue as a major source of inflammatory cytokines to determine the gene expression of IL-6, MCP-1, and TNFα. Nonetheless, we did not observe a difference between the groups in IL-6 gene expression (Figure 6D). Regarding the other two inflammatory cytokines, TNFα showed a significant difference between the STD and HFD and the HFD+A groups (*p* < 0.05) (Figure 6E) and the HFD group had a higher gene expression of MCP-1 significantly different from that of the STD (*p* < 0.01) and STD+A (*p* < 0.05) groups (Figure 6F). However, alliin treatment did not show any detected effect in the gene expression of inflammatory cytokines.

### 3.5. The mRNA Expression of Antioxidant Enzymes in the Liver Was Not Modified by Alliin

Oxidative stress has been strongly linked to metabolic dysfunction in obesity, implicated in the development of type 2 diabetes mellitus and nonalcoholic fatty liver disease [20,21,22]. Endogenously, the liver is the major producer of antioxidant enzymes like glutathione peroxidase (GPx), superoxide dismutase-1 (SOD-1), and catalase to balance the activity of reactive oxygen and nitrogen species [23].

The antioxidant effect of nutraceuticals is usually attributed to their interactions with free radicals or to their capacity to induce endogenous antioxidant enzymes [24]. Moreover, the antioxidant activity of garlic has already been reported [25], but alliin had only a few reports regarding this matter [26,27]. That being the case, we decided to determine the ability of alliin to induce the expression of three endogenous antioxidant enzymes in the liver. The mRNA expression of antioxidant enzymes in the liver was not modified by the diet or the alliin treatment (Figure 7).

## 4. Discussion

The caloric imbalance produced by physical inactivity and the excessive caloric and lipidic content in western diets are the main etiology of obesity in humans [28]. The DIO mice model is characterized by the 60% of caloric content from lipids for at least eight weeks [29,30], that allows it to mimic the main etiology, physiopathology, and comorbidities associated with human obesity. Although the specific criteria and metabolic levels to establish the DIO model are not fully standardized [30], for the present study, we considered as hallmarks for the establishment of obesity the differences in body weight and in the epididymal adipose tissue, as well as triglyceride levels, glucose tolerance, and insulin resistance through the OGTT and IITT. In this case, we administrated the high-fat diet for nine weeks. As shown in Figure 2, body weight was successfully increased by the diet. Other parameters such as cholesterol, triglycerides, and glucose levels were also affected by the increased lipid intake (Appendix A). Once obesity was established, we proceeded with the alliin treatment. 

### 4.1. Alliin Reduces the Adipocyte Size and Improves Metabolism Leading to Homeostasis of Adipose Tissue

After 3.5 weeks of daily alliin administration, the main effect was the general diminishment of glucose levels, reflected in the area under the curve of OGTT. Specific points in the oral glucose tolerance test, intraperitoneal insulin sensitivity test, and parameters like triglycerides and cholesterol concentrations were not markedly affected, in contrast to the findings of Zhai et. al. (2018), where alliin proved its hypoglycemic and hypolipidemic effects [31]. As a probable explanation, the major difference between these reports was the time of administration (3.5 weeks vs. 8 weeks), which could explain that over a shorter period of administration, alliin still does not fully exert its hypoglycemic and hypolipidemic effects. Remarkably, it can reduce glucose levels in this short period. In general, garlic and others of its compounds have been widely proven to be hypoglycemics and hypolipidemics [32,33,34,35]. 

Additionally, alliin reduced the size of the adipocytes, suggesting its implication in lipid metabolism by modifying its storage [36], by inhibiting lipid synthesis, or by affecting acetate incorporation into fatty acid [37,38]. Interestingly, we did not find any obvious hypolipidemic effect, despite its being widely reported [35,39] even in previous DIO models where a significant decrease of lipids was found. Again, the main differences compared with our design were the time and dosage form [31]. This reduction in the adipocyte size could constitute an initial step in the further reduction of lipid storage and/or inhibition synthesis exhibited when it is administrated for longer periods.

The hypertrophy phenomenon of the adipose tissue favors cellular death, tissue hypoxia, basal lipolysis, and the liberation of free fatty acids, resulting in an elevated inflammatory cytokine expression and secretion crosslinked to signaling pathways culminating in insulin resistance [40,41]. In this sense, decreasing the adipocyte size implies a metabolic improvement and stress reduction in the tissue [19,42]. Additionally, reports of the antihypertrophic effect of garlic compounds in cardiomyocytes ascribe this property to its antioxidant effect and the ERK pathway participation [43,44].

In adipose tissue, the antiobesity effects of nutraceuticals have been studied for their capacity to influence lipid absorption, energy intake, and lipogenesis, by their ability to increase lipolysis and energy expenditure, as well as their effect on adipocyte differentiation and proliferation [44]. Even though alliin´s mechanism of action remains unknown, evidence points to the activation of PPARγ in adipose tissue [16,45]. PPARγ activators have been used as type 2 diabetes mellitus treatment, acting as insulin sensitizers but with important secondary effects like increased lipogenesis or lipogenic intake in the liver, leading to steatosis [46,47,48]. This mechanism could explain the tendency of TG levels (Table 1, Figure 3) and the generation of initial liver steatohepatitis found in the STD+A group (unpublished data). Additionally, the effect observed in the inflammatory cytokine IL-6 and the adipocytokines leptin and resistin could be interfering in the hypoglycemic effect and the decreased adipocyte hypertrophy.

### 4.2. Alliin Showed a Relevant Anti-Inflammatory Potential in the Low-Grade Inflammatory State in Obesity 

The effect of garlic and its organosulfur compounds on immunological improvement has been previously described [14,15,16,49]; albeit, these properties have been mostly attributed to its antioxidant capacity [11,13,50,51]. Approaching obesity as an inflammatory condition, and due to the limited research on alliin on inflammatory parameters, we determined the gene expression and secretory levels of inflammatory cytokines and adipocytokines after treatment with alliin in DIO mice.

The adipocytokines resistin and leptin are tightly related to adipose tissue [52] and both were clearly affected by alliin. Their gene expression and secreted levels were evidently reduced by alliin treatment, which can also explain the effect observed in the decrease of adipocyte size and signals the influence on metabolic parameters, enabling the metabolic and immune homeostasis [42].

Resistin is a cysteine-rich protein [53,54] whose main role has been described in the development of insulin resistance, inflammation, and endothelial alterations in cardiovascular diseases [53,55]. Although its receptor is not fully identified, the literature describes that it could perform its functions through TLR4 [56,57]. On the other hand, it has been reported that it can block the interaction of LPS with TLR4 and avoid endotoxic shock [58], which would indicate a primary biological function in regulating inflammation.

That the gene expression and serum levels of resistin diminished with alliin treatment is an interesting finding, in concordance with the microarray analysis by our team in 3T3-L1 adipocytes [15], where it was proposed that alliin could be interacting with the TLR4 receptor, blocking its signaling and decreasing inflammatory cytokine secretion. In this sense, it is feasible to propose that alliin competes in some way with resistin for the receptor, regarding its function in mitigating inflammation, affecting the function and regulation of the resistin itself.

It is known that leptin levels are tightly related to the metabolic state of adipose tissue [52], which means that an increased need for lipid storage produces adipocyte hypertrophy and these, in turn, secrete leptin to induce lipolysis for the regress of metabolic stress [59,60]. Additionally, its satiety function in the hypothalamus is attenuated, because of the incorrect ability of leptin to suppress appetite, named leptin resistance, promoting energy intake by the subject [61]. The fact that alliin significantly decreased leptin levels explains the improvement of satiety and implies an improvement in lipid metabolism by diminishing adipocyte metabolic stress, which involves the functions described for leptin [62,63]. In addition, because adiponectin is considered a functional leptin antagonist, subsequently, the reduction of leptin suggests a possible improvement in adiponectin’s activity: insulin sensitivity, adequate adipogenesis, and amelioration of hyperlipidemia showing lipid homeostasis [59].

Inflammatory cytokines like IL-6, TNFα, and MCP-1 are also described as acute-phase proteins, and are the first mediators of local acute inflammatory processes; upon reaching circulation they may generate systemic inflammation [64] and their prolonged secretion determines a chronic inflammatory state [65]. Alliin has demonstrated its anti-inflammatory effect by diminishing the concentrations of circulating cytokines, like IL-6, TNFα, and MCP-1, in several inflammatory disease models [15,16,46]. In those models, however, the inflammatory process is generated by an acute and strong LPS insult, while obesity is characterized by a low-grade inflammatory state, hence, by definition, lower levels of inflammatory cytokines can be expected [6].

We observed the immunomodulatory effect of alliin in the decrease of gene expression and secretion of IL-6 and in the tendency to homogenize the levels of TNFα and MCP-1, interestingly only in the HFD+A and not the STD+A group. This suggests an improvement in adipose tissue homeostasis by decreasing the major role of IL-6 in macrophage M1 polarization [66] and the secretion of acute-phase cytokines, like TNFα and CRP, and therefore reducing the low-grade inflammatory state [67,68,69].

The lack of significant differences in the gene expression of inflammatory cytokines indicates that alliin’s effect over serum concentration of IL-6 and MCP-1 and TNFα could be the result of a post-translational regulation or, at least, at the post-transcriptional level. There are widely recognized crisscrossed signaling networks between inflammatory cytokines and metabolic signaling pathways [6,18] and a therapeutic approach to one or another generally involves the enhancement of metaflammation.

Finally, we have shown that main liver antioxidant enzymes were not modified by alliin or the different diets; this probably means that the insult and the time were not enough to produce a detectable effect on the antioxidant enzymes’ gene expression. 

## 5. Conclusions

In summary, alliin demonstrated a clear effect on certain metabolic parameters such as improving glucose tolerance and decreasing adipocyte hypertrophy. Additionally, it influenced the decrease in serum leptin and resistin levels and it favored homeostasis, which in turn was influenced by a less inflammatory environment reflected by the decrease in IL-6 and the stabilization of TNFα and MCP-1 serum levels.

## Figures and Tables

**Figure 1 nutrients-12-00624-f001:**
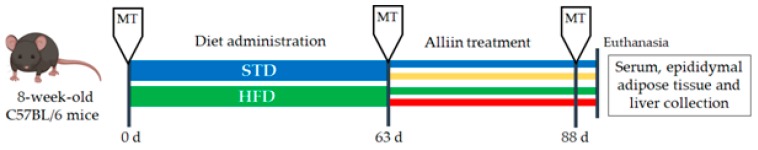
Study design. Metabolic test (MT) that included cholesterol determination, triglycerides test, Oral Glucose Tolerance Test (OGTT), and Intraperitoneal Insulin Tolerance Test (IITT) were performed three times during the study: basal, post-diet treatment, and post-alliin treatment. Once the alliin treatment was finished, euthanasia and tissue collection were performed.

**Figure 2 nutrients-12-00624-f002:**
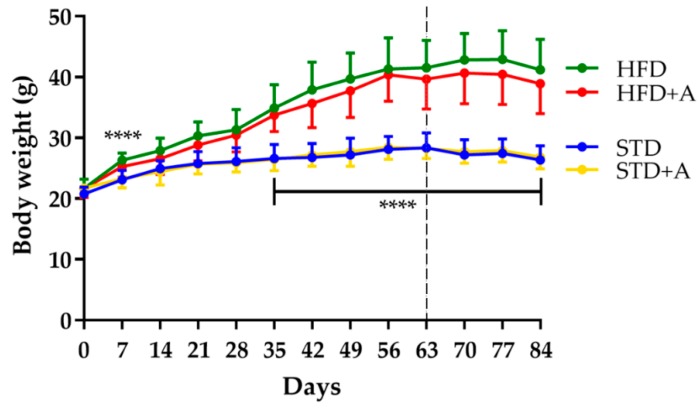
Body weight definitive differences between diet groups were established until day 35, **** *p* < 0.0001 by one-way ANOVA/Tukey’s test. Alliin treatment (day 63, slashed line) did not have an effect on body weight. STD, standard diet group; STD+A, standard diet + alliin treatment group; HFD, high-fat diet group; HFD+A, high-fat diet + alliin treatment group.

**Figure 3 nutrients-12-00624-f003:**
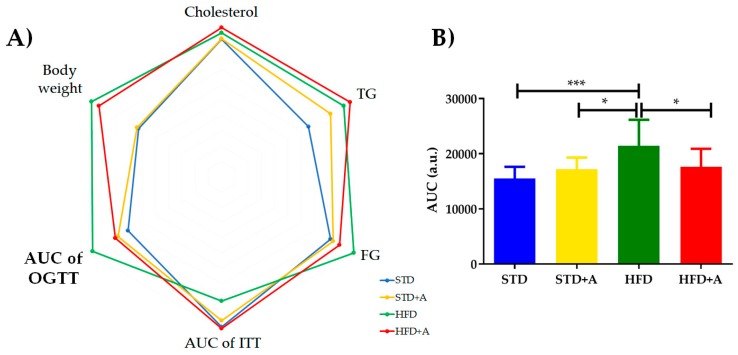
(**A**) Polygonal plot shows the normalized values obtained in the metabolic tests according to the group. (**B**) AUC of OGTT showed a statistically significant difference between HFD and the other three groups (vs. STD *** *p* < 0.001, and vs. HFD+A and STD+A groups **p* < 0.05), expressed as mean ± SD. TG: triglycerides, FG: fasting glucose, AUC of OGTT: area under the curve of oral glucose tolerance test, AUC of ITT: area under the curve of insulin tolerance test. STD, standard diet group; STD+A, standard diet + alliin treatment group; HFD, high-fat diet group; HFD+A, high-fat diet + alliin treatment group.

**Figure 4 nutrients-12-00624-f004:**
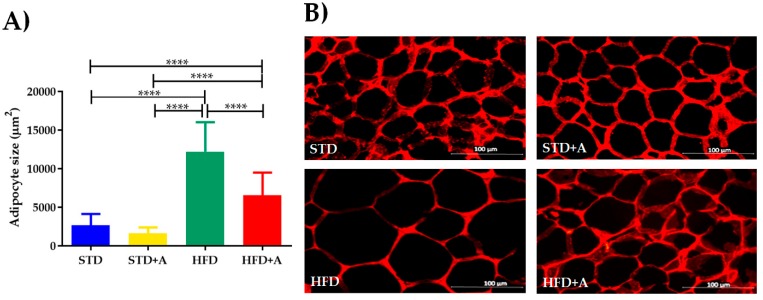
(**A**) Adipocyte size, differences observed between groups on different diets. HFD+A group showed significantly decreased adipocyte size compared to HFD with a significant *p* value (*****p* < 0.0001) via Kruskal–Wallis/Dunn’s test, presented as mean ± SD. (**B**) Representative image of adipocyte size per group. STD, standard diet group; STD+A, standard diet + alliin treatment group; HFD, high-fat diet group; HFD+A, high-fat diet + alliin treatment group.

**Figure 5 nutrients-12-00624-f005:**
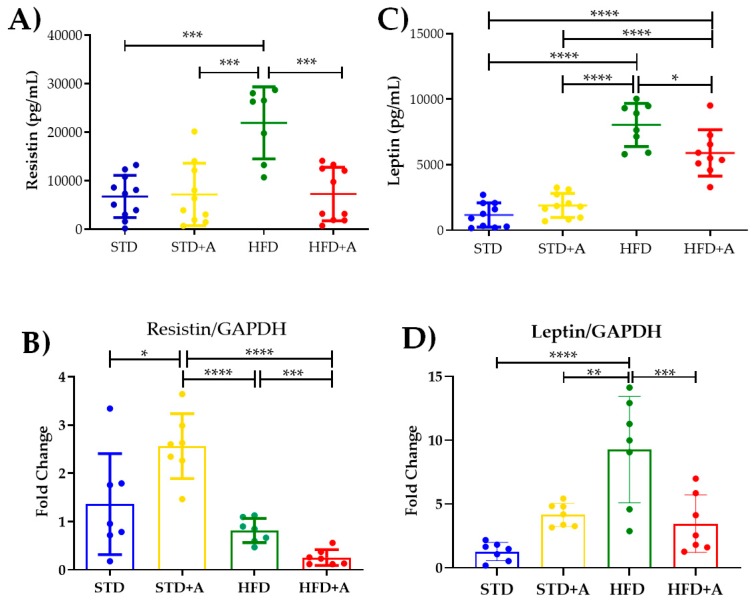
**Effect on secretion and expression of adipocytokines.** (**A**) The effect of alliin in resistin secretion was notably significant between the HFD and the HFD+A groups (****p* < 0.001). (**B**) This difference was also observed in the gene expression of resistin (****p* < 0.001). (**C**) Similar behavior was shown in leptin serum levels in the HFD+A group (**p* < 0.05), (**D**) a difference that persisted in the gene expression of leptin (** *p <* 0.01, ****p* < 0.001, *****p* < 0.0001). STD, standard diet group; STD+A, standard diet + alliin treatment group; HFD, high-fat diet group; HFD+A, high-fat diet + alliin treatment group. Adipocytokine serum levels were analyzed by one-way ANOVA/Tukey’s test. For gene expression, each sample 2^-∆∆CT^ was calculated and analyzed by one-way ANOVA/Tukey’s test, except for resistin gene expression, where Student’s T test was performed. Values shown as mean ± SD.

**Figure 6 nutrients-12-00624-f006:**
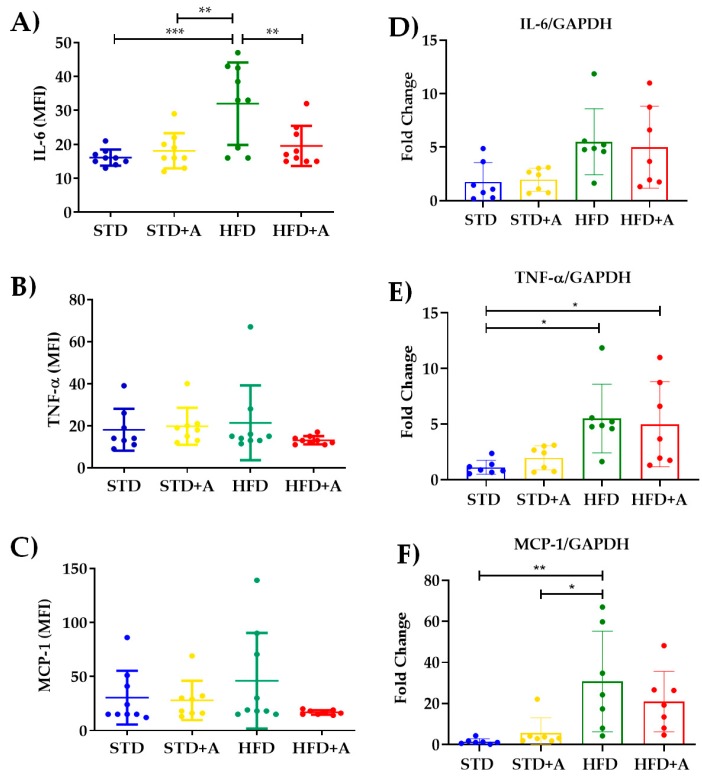
**Alliin effect on proinflammatory cytokines.** (**A**) Alliin demonstrated its anti-inflammatory capacity by decreasing the levels of IL-6 in the HFD+A group in a significant manner (***p* < 0.01); (**B**) TNFα and (**C**) MCP-1 levels. (**D**) IL-6 gene expression did not have any difference, and (**E**) TNFα had differences between the STD and HFD and HFD+A groups, and a similar result was seen in (**F**) MCP-1 with differences in HFD vs. STD and STD+A groups. * *p* < 0.05, *** *p* < 0.001. STD, standard diet group; STD+A, standard diet + alliin treatment group; HFD, high-fat diet group; HFD+A, high-fat diet + alliin treatment group. For gene expression, each sample 2^-∆∆CT^ was calculated and analyzed by one-way ANOVA/Tukey’s test. One-way ANOVA/Tukey’s test was also performed for cytokine serum levels. Values shown as mean ± SD.

**Figure 7 nutrients-12-00624-f007:**
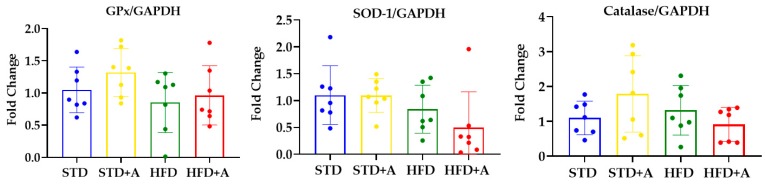
**Alliin effect on antioxidant enzymes gene expression.** Glutathione peroxidase, superoxide dismutase, and catalase gene expression in the liver of DIO mice was not modified by alliin treatment. STD, standard diet group; STD+A, standard diet + alliin treatment group; HFD, high-fat diet group; HFD+A, high-fat diet + alliin treatment group. For gene expression, each sample 2^-∆∆CT^ was calculated and analyzed by one-way ANOVA/Tukey’s test. Results expressed as mean ± SD.

**Table 1 nutrients-12-00624-t001:** Primers used for qRT-PCR.

Gene	Forward	Reverse	Product	Temperature
*Leptin*	GTCTTATGTTCAAGCAGTGCC	TGAAGCCCAGGAATGAAGT	150	58 °C
*Resistin*	CCCACGGGATGAAGAACCTTT	CACGCTCACTTCCCCGACAT	372	63 °C
*MCP-1*	GGTGTCCCAAAGAAGCTGTAG	CTGAAGACCTTAGGGCAGATG	161	61 °C
*TNFα*	GCTGAGCTCAAACCCTGGTA	CGGACTCCGCAAAGTCTAAG	118	63 °C
*Catalase*	GCAGATACCTGTGAACTGTC	GTAGAATGTCCGCACCTGAG	229	59 °C
*GPX*	GTCCACCGTGTATGCCTTCT	TCTGCAGATCGTTCATCTCG	152	60 °C
*SOD1*	TGGTGGTCCATGAGAAACAA	GTTTACTGCGCAATCCCAAT	115	61 °C
*GAPDH*	TCCACCACCCTGTTGCTGTA	ACCACAGTCCATGCCATCAC	452	63 °C

**Table 2 nutrients-12-00624-t002:** Effect of alliin on metabolic tests.

	STD	STD+A	HFD	HFD+A
Cholesterol (mg/dL)	152.75 ± 3.19	153.36 ± 2.91	159.50 ± 5.19	165.64 ± 14.71
Triglycerides (mg/dL)	104.40 ± 14.78	131.20 ± 22.31	147.30 ± 36.29	155.00 ± 33.97
Fasting glucose (mg/dL)	109.20 ± 17.62	112.20 ± 14.31	132.70 ± 27.04	118.50 ± 27.86
AUC of ITT	11,014 ± 1512	10,549 ± 1,212	9,119 ± 1,723	11,134 ± 4,977
**AUC of OGTT ***	15,506 ± 2124	17,220 ± 2,099	**21,458 ± 4,697 ***	**17,618 ± 3,285 ***
Body weight (g)	26.33 ± 2.34	26.81 ± 1.94	41.20 ± 5.00	38.90 ± 4.94
Epididymal adipose tissue weight (g)	0.66 ± 0.14	0.78 ± 0.067	2.65 ± 0.60	2.48 ± 0.42

*A statistically significant difference was found in the area under the curve (AUC) of the oral glucose tolerance test (OGTT) between the HFD and HFD+A groups (**p* < 0.05). STD, standard diet group; STD+A, standard diet + alliin treatment group; HFD, high-fat diet group; HFD+A, high-fat diet + alliin treatment group. Cholesterol determination was analyzed by Kruskal–Wallis/Dunn’s test. Triglycerides, fasting glucose, each time in the OGTT and Intraperitoneal Insulin Tolerance Test (IITT) AUC of both tests, body weight, and epididymal adipose tissue weight were analyzed by one-way ANOVA/Tukey’s test. The values are expressed as mean ± SD.

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
