# Peer review of "Alliin, An Allium sativum Nutraceutical, Reduces Metaflammation Markers in DIO Mice"

_nutrients, 2020, doi:10.3390/nu12030624_

Round 1
Reviewer 1 Report
The manuscript entitled “Alliin, an Allium sativum nutraceutical, reduces metaflammation markers in DIO mice” submitted for revision in the Nutrients had been positively reviewed with some minor modifications.
This is an interesting manuscript concerning the effect of alliin supplementation on metainflammatory markers for energy metabolism, antioxidant enzymes and proinflammatory cytokines in diet-induced obesity (DIO) mice.
Research results expand knowledge of Allium Sativum and alliin. The Authors described the potential effect allium on certain metabolic parameters such as improving glucose tolerance and decreased adipocyte hypertrophy. According to the Authors allium influences the decrease in serum leptin and resistin levels, that it favors homeostasis, which in turn is influenced by a less inflammatory environment reflected by the decrease in IL-6 and the stabilization of TNFα and MCP-1 serum levels. This is valuable information about the importance of garlic in nutrition and uses in food production.
The review is positive. Therefore, I propose minor amendments.
The abstract should be changed: the purpose of the study and a brief description of the experiment should be added. No consistency in the length of experience. The authors write in the abstract: “In the present work we have demonstrated that treatment with alliin for 3.5 weeks in DIO mice…” The experiment should be described in more detail. Many abbreviations and names of enzymes, markers are used in the text. I suggest adding a list of abbreviations used at the beginning of the publication. Keywords: I suggest you delete the words “Metaflammation”, “Alliin”. These words are already in the title of the publication and should not be repeated The captions under figures should be unified. Sometimes it is given **** p < 0.0001, and sometimes only p<0.0001 (without ****) ex. Figure 2 and Figure 4A. Figure 3. The authors write:”statistically significant difference between HFD and HFD+A groups (p < 0.05)…” but not information about the statistically significant difference between STD+A - HFD and also HFD –HFD+A. The same applies to the remaining figures. There is no description of all significant differences. The captions under figures 3 (B), 4 (A), 5 should contain information about significant differences in all data.
Author Response
Reviewer #1
- The abstract should be changed: the purpose of the study and a brief description of the experiment should be added. No consistency in the length of experience.
RESPONSE: the purpose of the study and a brief description of the experiment has been included in the abstract as follows: “After 9 weeks of HFD and once established obesity in C57BL/6 mice, we observed that the daily treatment… both evaluated through a blood glucose monitoring system. Once alliin treatment was completed, serum, adipose tissue and organs of interest related to metabolism were removed for further analysis… evaluated with microscopy... were detected by qRT-PCR and ELISA, respectively.”
- The authors write in the abstract: “In the present work we have demonstrated that treatment with alliin for 3.5 weeks in DIO mice…” The experiment should be described in more detail
RESPONSE: Done as previously indicated.
- Many abbreviations and names of enzymes, markers are used in the text. I suggest adding a list of abbreviations used at the beginning of the publication.
RESPONSE: a list of abbreviations has been added at the beginning of the manuscript.
- Keywords: I suggest you delete the words “Metaflammation”, “Alliin”. These words are already in the title of the publication and should not be repeated
RESPONSE: Done
- The captions under figures should be unified. Sometimes it is given **** p < 0.0001, and sometimes only p<0.0001 (without ****) ex. Figure 2 and Figure 4A. Figure 3.
RESPONSE: Done
- The authors write:”statistically significant difference between HFD and HFD+A groups (p < 0.05)…” but not information about the statistically significant difference between STD+A - HFD and also HFD –HFD+A.
RESPONSE: Information has been added where corresponds
- The same applies to the remaining figures. There is no description of all significant differences.
RESPONSE: Done as previously mentioned
- The captions under figures 3 (B), 4 (A), 5 should contain information about significant differences in all data.
RESPONSE: Information about significance has been included in the text where it corresponds. (marked in red in the resubmitted text)

Reviewer 2 Report
Line 32, DIO - explain
Line 58 thence, should be hence
Line 95 …STD+A who were treated with alliin (15 mg/kg) besides the standard diet (15 mg/kg)…. Twice 15 mg/kg was used
Line 97 Euthanasia procedure is missing, please add.
Line 143 The ANOVA one or two way? Please specify
Line 144 ….was be determined… should be was determined
Fig. 5c was the difference significant STD+A - HFD? - in my opinion it looks as it is significant
the same for fig 5b; was the difference significant STD+A - HFD?
And The same for Fig 6.
And also please add the description of the significance *, **, *** in the Figures captions. More information is needed in Fig cations.
Author Response
Reviewer #2
- Line 32, DIO – explain
RESPONSE: Done
- Line 58 thence, should be hence
RESPONSE: Done
- Line 95 …STD+A who were treated with alliin (15 mg/kg) besides the standard diet (15 mg/kg)…. Twice 15 mg/kg was used
RESPONSE: Corrected, duplicated has been erased.
- Line 97 Euthanasia procedure is missing, please add.
RESPONSE: A description has been added as follows: “Finally, they were euthanized by decapitation with scissors for whole blood collection and for tissue dissection.”
- Line 143 The ANOVA one or two way? Please specify
RESPONSE: Done, it was One-way ANOVA, as now indicated.
- Line 144 ….was be determined… should be was determined
RESPONSE: Done
- 5c was the difference significant STD+A - HFD? - in my opinion it looks as it is significant
RESPONSE: We thank the reviewer to point out this omission. Consequently, we corrected this by adding the missing statistically significance. See new figure 5.
- the same for fig 5b; was the difference significant STD+A - HFD?
RESPONSE: Again, we thank the reviewer for his/her comment on this. In this case we performed the analysis by using Student´s T-test, therefore the statistically significances presented correspond to one-by-one comparisons among the groups.
- And the same for Fig 6.
RESPONSE: In this figure we indicate only the comparisons where a statistically significant differences were obtained.
- And also please add the description of the significance *, **, *** in the Figures captions. More information is needed in Fig cations.
RESPONSE: The descriptions has been added to the figure legends.
